Accumulation and transfer of polystyrene microplastics in Solanum nigrum seedlings

Zhang JuKui
Cao Lian
Zhu Xiaoyan
Li Hanbo
Duan Gang
Wang Ying wangying19879@aliyun.com
Key Laboratory of Songliao Aquatic Environment, Ministry of Education, Jilin Jianzhu University , Changchun , China
Landa Blanca
Electronic publication date: 2023 Aug 31
Publication date: 2023
Volume: 11
Electronic Location ID: e15967
Received 2023 Mar 7; Accepted 2023 Aug 4
Copyright: ©2023 Zhang et al.
Copyright year: 2023
Copyright holder: Zhang et al.
License: This is an open access article distributed under the terms of the Creative Commons Attribution License, which permits unrestricted use, distribution, reproduction and adaptation in any medium and for any purpose provided that it is properly attributed. For attribution, the original author(s), title, publication source (PeerJ) and either DOI or URL of the article must be cited.
License URL: https://creativecommons.org/licenses/by/4.0/

Keywords: Accumulation, Microplastic, Uptake, Transfer, Solanum nigrum

Funding: Natural Science Foundation of Jilin Province YDZJ202201ZYTS480 National Water Pollution Control and Management Technology Major Project of China 2012ZX07201–001 Strategic Priority Research Program of the Chinese Academy of Sciences XDA28020400 This work was supported by the Natural Science Foundation of Jilin Province (YDZJ202201ZYTS480), the National Water Pollution Control and Management Technology Major Project of China (2012ZX07201–001), and the Strategic Priority Research Program of the Chinese Academy of Sciences (XDA28020400). The funders had no role in study design, data collection and analysis, decision to publish, or preparation of the manuscript.

==============================
Microplastic (MP) pollution is lately receiving increasing attention owing to its harmful impact on terrestrial ecosystems. In this microcosm study, we assessed the uptake and transfer of MPs in Solanum nigrum seedlings exposed to 50 mg L–1 of 0.2-µm polystyrene (PS) beads for 30 d. Confocal laser scanning micrographs helped detect highly intense red fluorescence signals from PS-MP beads in S. nigrum root compared with the controls. Confocal images revealed that the PS beads were primarily distributed in the epidermis and xylem of roots and vascular systems of stems and leaves. Scanning electron microscopy showed that PS beads were scattered on the cell walls of the root xylem and leaf vascular system. Few PS beads were transferred from roots to stems and leaves via the vascular system following the transpiration stream. In conclusion, our findings showed that PS beads accumulated in S. nigrum roots and were transferred from the roots to the aerial parts.

Introduction

Microplastics (MPs) are an emerging pollutant, receiving increasing attention in recent years (Santillo, Miller & Johnston, 2017; Luo et al., 2022; Krishnamoorthi et al., 2022). It is estimated that annually 4 to 23 times more plastic is released into the terrestrial environment than the marine environment (Horton et al., 2017). Previous studies have shown that a large amount of MPs release into the terrestrial environment from wastewater irrigation, sludge utilization, household waste, mulching, and other sources (Bläsing & Amelung , 2018). Plastics continue entering the environment due to huge production levels, widespread consumption, and poor waste management. The soil-plant system further aggravates the risks of MP accumulation in the soil (Wang et al., 2022a; Wang et al., 2022b; Zhou et al., 2022). Nano-plastic (NP) particles can be absorbed by plants and animals, leading to their transfer to humans through the food chain, adversely affecting human health (Diepens & Koelmans, 2018; Wang et al., 2022c).

The chemical structure of plastics makes them resistant to degradation, leading to the accumulation of a large amount of plastic waste in the environment, especially in soil and sediment and other poor mobility of environmental media (Rodrigues et al., 2018; Xu et al., 2019). The adsorption of pollutants on the surface of MPs promotes the migration and transformation of other pollutants in the soil with the migration of MPs (Liu et al., 2019). Meanwhile, extremely small-sized MPs can be absorbed by plant and animal cells via the transpiration stream, which has further raised concerns about the danger and risks they pose to the environment (Lambert & Wagner, 2016; Prata et al., 2020).

Many researchers have focused on the effects of MPs on plant growth and reproduction (Bosker et al., 2019; Machado et al., 2019; Bouaicha et al., 2022). However, only a few studies highlighted the uptake and transfer of MPs via plants (Li et al., 2021; Gao et al., 2023; Li et al., 2023a). Given their extremely small size, NPs can enter the roots, stems, and leaves of plants (Giorgetti et al., 2020; Jiang et al., 2022). A previous study demonstrated that BY-2 cells could uptake 20- and 40-nm nano-polystyrene beads via endocytosis suggesting that small-sized MPs (<100 nm) can enter tobacco cells (Bandmann et al., 2012). Zhou et al. (2021), using confocal laser scanning micrographs, confirmed the uptake of 20-nm polystyrene (PS)-NPs by rice roots upon exposure of rice seedlings to PS-NPs for 16 d. Li et al. (2020a) also showed that wheat roots could uptake small-sized PS-MPs and transfer them to stems and leaves via the vascular system.

Accumulation of MPs in terrestrial plants increases the risk of direct exposure of humans to toxic contaminants. Several studies have presented evidence of the absorption of MPs by plants (Li et al., 2020b; Li et al., 2023b. Moreover, NPs have been shown to slowly provide typical micronutrients (Spielman-Sun et al., 2018) and induce the accumulation of heavy metal (Wang, Wang & Song, 2021). MP pollution in the terrestrial ecosystem is a potential threat to plants and, consequently, to human health. S. nigrum is an annual or perennial plant and typical cadmium (Cd)-hyperaccumulator. This species is used toremediate contaminated soil, playing a major role in restoring heavy metal-contaminated soil (Liu et al., 2015; Li et al., 2019a). To the best of our knowledge, the accumulation and transfer of MPs in this Cd-hyperaccumulator species have not been studied yet. In this microcosm study, we assessed the uptake and accumulation of MPs by S. nigrum roots, followed by the transfer of MPs to stems and leaves. Therefore, investigating the accumulation of MPs in S. solanum can help improve its efficiency in remediating polluted land.

Materials and Methods

Materials

Stock solutions of PS beads labeled with Nile blue (NB) and 4-chloro-7-nitro-1, 2, 3-benzoxadiazole (NBD-Cl) were purchased from Tianjin Da’e Scientific Co., Ltd. (China). We used 0.2-µm red and green fluorescent polystyrene (PS) beads with high-intensity fluorescence at excitation/emission wavelengths of 635/680 nm and 480/515 nm, respectively. The microstructure of PS beads, which were all spherical, was examined using environmental scanning electron microscopy (SEM: FEI Quanta 450 FEG FESEM) under high vacuum conditions and the PS beads appeared regular sphericalundera transmission electron microscope (TEM, HT7800) (Fig. 1).

Figure 1 Microstructure of PS beads (stock solutions) under SEM (A) and transmission electron microscopy (B).

Plant materials and culture

The study was conducted at Jilin Jianzhu University in 2022, Jilin Province, China. S. nigrum seeds were collected randomly from the fields around Changchun city and stored at 4 °C for germination and seedling culture. The seeds were sterilized with 10% NaClO solution and washed thrice with distilled water. The treated seeds were placed on a Petri dish (90 mm) with filter paper and then transferred to a growth chamber. The chamber was maintained at 30–20 °C with a 12-h photoperiod for 5 d.

The stock solutions of PS beads (two mL) labeled with NB and NBD-Cl were dispersed using ultrasonic vibrations for 3 min. They were then mixed with 400 mL Hoagland nutrient solution. This nutrient solution comprised 5 mM Ca2+, 2.00 mM Mg2+, 6.04 mM K+, 2. 22 × 10−2 mM EDTA-Fe2+, 6.72 × 10−3 mM Mn2+, 3.16 × 10−4mM Cu2+, 7.65 × 10−4 mM Zn2+, 2.10 mM SO42−, 1 mM H2PO4−, 4.63 × 10−2 mM H3BO3, 5.56 × 10−4 mM H2MoO4, and 15.04 mM NO3−. The mixed solution was again dispersed for 3 min using ultrasonic vibrations. Two seedlings, cultured for 5 d, were transferred to a non-transparent beaker containing 400 mL mixed solution. The beaker was then covered with a cap while keeping the shoots of the seedlings outside the beaker through a small hole in the cap (Li et al., 2023b). Seedlings of uniform size were exposed to 0 (absence of plastic beads) and 0.2 µm PS beads labeled with NB and NBD-Clas 50 mg L−l solutions for 30 d. Three replicates were used for each treatment in each experiment. Another three replicates of PS beads labeled with NB were used for each treatment to determine plant height and biomass. The seedlings were cultured in the mixed solution for another 30 d in the growth chamber under the conditions described above.

Absorption and transfer of PS beads

Plant height and biomass

All the seedlings were harvested after culturing for 30 d. Plant heights were determined manually using a ruler. Then, the shoot and root parts of the seedlings were separated and put into separate envelopes. The separated samples were dried at 60 °C for 48 h to determine the dry weight.

Confocal microscopy

The primary roots of the seedlings cultured for 30 d (two cm from the base of the root), exposed to PS with NB, were harvested and cleaned with pure water to remove the PS beads attached to the root surface. The stems (middle of the stem region) and the leaves (the third blade from the bottom) exposed to PS with NBD-Cl were also harvested. Subsequently, the samples were embedded in 4% low melting point agarose. The roots, stems, and leaves were sectioned into 80, 60, and 60 µm sections, respectively, using a vibrating microtome (VT1200S; Leica). The sectioned samples were placed on the glass slide. A few drops of phosphate-buffered saline (PBS) were added to keep the section hydrated. Finally, the specimen was covered with a coverslip. Then, a confocal laser scanning microscope (CLSM, Zeiss LSM 880) at the excitation/emission wavelengths of 635/680 nm and 480/515 nm was used to examine the sectioned tissues (structure diagram is presented in Fig. S1). Transverse and longitudinal sections of the roots, stems, and leaves were examined, and three replicates were assessed. A built-in SPOT camera was used to capture the images, which were imported into microscope imaging software (ZEN 2.6 blue edition) to adjust for brightness and contrast. The method of collecting confocal images as previously described in Li et al. (2020a); Li et al. (2020b) was modified and used in this study. Specifically, the exposure time, sampling site and section thickness were improved.

Scanning electron microscopy

The roots and leaves were washed with distilled water and the samples were cut into 0.5 cm sections, which were then frozen immediately in liquid nitrogen for 1 d. Then the SEM images were obtained according to the method of Li et al. (2023b). The frozen sections were dried by a lyophilizer (FD-1A-80 BIOCOOL) and coated with a 1-nm thick gold layer using a sputter coater (Cressington model 108; Ted Pella). At last, an SEM was used to examine the presence of PS beads at high vacuum mode with backscatter and an accelerated potential of 20 kV. Images of the cross-sections were captured at different magnifications. The images of the xylem vessel of the roots and leaves were magnified, where high levels of PS beads were accumulated in all the treatments. Three replicates were used for all treatment groups.

Data analysis

One-way ANOVA was used to analyze the variation in plant height, shoot and root biomass under different treatments. Statistically significant differences between treatments were analyzed using the Least Significant Difference (LSD) test, with a critical significance level of P = 0.05. Statistical analysis were carried out using the SPSS software (Version 17.0, SPSS Inc., Chicago IL, USA).

Results

Plant height and biomass

The mean plant heights were 15.6 cm and 15.0 cm after treatment with 0 and 0.2 µm MPs, respectively. The presence of PS beads had no significant effect on plant height and root and shoot biomass of S. nigrum seedlings after exposure to PS beads for 30 d (Table 1).

Table 1 Effect of exposure to NB-labeled PS-MPs beads on the plant height, shoot and root biomass of S. nigrum.

Three replicates were used for all treatment groups.

Treatment (µm)	0	0.2	
Plant height (cm)	15.6 ±0.44a	15.0 ±0.50a	
Shoot biomass (g)	0.34 ±0.009a	0.31 ±0.009a	
Root biomass (g)	0.12 ±0.002a	0.12 ±0.007a	
Notes.

Within lines, means followed by the same letter are not significantly different at P = 0.05.

Uptake of PS beads by S. nigrum roots

Red fluorescence signals were detected for two exposure groups of 0.2-µm PS beads at 635 nm using the CLSM. CLSM analysis showed stronger luminescence signals from PS beads in the transverse and longitudinal sections of the seedlings cultured with 50 mg L−l PS beads compared to the seedlings cultured in the absence of PS beads. A substantial proportion of PS beads were primarily distributed in the epidermis and xylem tissues (Figs. 2 and 3). CLSM images of root sections did not reveal any PS beads in the cortex tissues (Figs. 2D– 2I and 3D–3I).

Figure 2 Confocal images of cross-sections of S. nigrum roots treated for 30 d with 50 mg L−1 solutions of 0.2-µm fluorescent NB-labeled polystyrene (PS) microbeads.

(A & D) Images of transverse sections of roots treated with CK and 0.2-µm NB-labeled PS-MPs beads in bright field, respectively. By adjusting the CLSM parameter to eliminate the interference of autofluorescence value in root tissues of control group. Under the same parameter, the fluorescence was observed in root tissues treated with 0.2 um PS-MPs beads. (B & E) The corresponding merged images in dark field, respectively. (C & F) The merged images of (A) and (B) and (D) and (E), respectively. (G–I) The local images of (D–F), respectively. Scale bars were 100 µm and three replicates were used for all treatment groups.

Figure 3 Confocal images of longitudinal sections of S. nigrum roots treated for 30 d with 50 mg L−1 solutions of 0.2-µm fluorescent NB-labeled polystyrene (PS) microbeads.

(A & D) Images of transverse root sections treated with CK and 0.2-µm NB-labeled PS-MPs in bright field, respectively. (B & E) The corresponding merged images in dark field. (C & F) The merged images of (A) and (B) and (D) and (E), respectively. (G–I) The local images of (D–F), respectively. By adjusting the CLSM parameter to eliminate the interference of autofluorescence value in root tissues of control group. Under the same parameter, the fluorescence was observed in root tissues treated with 0.2 um PS-MPs beads. Scale bars were 100 µm and three replicates were used for all treatment groups.

Transfer of PS beads from roots to aerial parts

To examine the transfer of PS beads from the roots to the aerial parts, green fluorescence PS beads were detected in stems and leaves using CLSM. Analysis of the stem sections revealed weak luminescence signals in the stem vasculature of the stem (Figs. 4D–4I and 5D–5I). The PS beads were primarily distributed in the leaf vasculature, as evident from the cross-section images of leaves (Figs. 6D–6I). In contrast, when cultured in the absence of PS beads, no fluorescence signals were detected in stems and leaves (Figs. 4A–4C, 5A-5C, and 6A–6C).

Figure 4 Confocal images of cross-sections of S. nigrum stems treated for 30 d with 50 mg L−1 solutions of 0.2-µm fluorescent NBD-Cl-labeled polystyrene (PS) microbeads.

(A & D) Images of transverse stem sections treated with CK and 0.2-µm NBD-Cl-labeled PS-MPs beads in bright field, respectively. (B & E) The corresponding merged images in dark field. (C & F) The merged images of (A) and (B) and (D) and (E), respectively. (G–I) The local images of (D–F), respectively. By adjusting the CLSM parameter to eliminate the interference of autofluorescence value in stem tissues of control group. Under the same parameter, the fluorescence was observed in stem tissues treated with 0.2 um PS-MPs beads. Scale bars were 100 µm and three replicates were used for all treatment groups.

Figure 5 Confocal images of longitudinal sections of S. nigrum stems treated for 30 d with 50 mL−1 solutions of 0.2-µm fluorescent NBD-Cl-labeled polystyrene (PS) microbeads.

(A & D) Images of longitudinal stem sections treated with CK and 02-µm NBD-Cl-labeled PS-MPs beads in bright field, respectively. (B & E) The corresponding merged images in dark field. (C & F) The merged images of (A) and (B) and (D) and (E), respectively. (G–I) The local images of (D–F), respectively. By adjusting the CLSM parameter to eliminate the interference of autofluorescence value in steam tissues of control group. Under the same parameter, the fluorescence was observed in stem tissues treated with 0.2 um PS-MPs beads. Scale bars were 100 µm and three replicates were used for all treatment groups.

Figure 6 Confocal images of cross-sections of S. nigrum leaves treated for 30 d with 50 mg L−1 solutions of 0.2-µm fluorescent NBD-Cl-labeled polystyrene (PS) microbeads.

(A & D) Images of transverse leaf sections treated with CK and 0.2-µm NBD-Cl-labeled PS-MPs beads in bright field, respectively. (B & E) The corresponding merged images in dark field. (C & F) The merged images of (A) and (B) and (D) and (E), respectively. (G–I) The local images of (D–F), respectively. By adjusting the CLSM parameter to eliminate the interference of autofluorescence value in leaf tissues of control group. Under the same parameter, the fluorescence was observed in leaf tissues treated with 0.2 um PS-MPs beads. Scale bars were 100 µm and three replicates were used for all treatment groups.

Moreover, SEM images clearly showed PS beads scattered on the cell wall of the xylem vessel in S. nigrum roots (Figs. 7A–7C). SEM images clearly showed dotted PS beads in leaf tissues (Figs. 7D–7F).

Figure 7 SEM images of the PS localization in S. nigrum roots and leaves treated for 30 d with 50 mg L−1 solution of 0.2-µm NB-labeled PS-MPs.

(A & B) (scale bar was 500 µm) PS-MPs beads in root xylem and leaf vasculature, respectively. (C & D) (scale bar was 20 µm) and (E & F) (scale bar was 1 µm), the enlarged images of (A) and (D), respectively. Three replicates were used for all treatment groups.

Discussion

Some researchers have revealed that MPs have a significant effect on plant growth. Exposure to PS-MPs has been shown to adversely affect the length of wheat plants (Liao et al., 2019). Wang, Wang & Song (2021) showed that combined polyethylene (1% and 10%, w/w) and Cd contamination inhibited the growth of lettuce (Lactuca sativa) plants and enhanced Cd enrichment in lettuce by increasing soil Cd bioavailability and soil-dissolved organic carbon content and decreasing soil pH. Weert et al. (2019) showed that root and shoot biomass of Myriophyllum spicatum and Elodea sp. positively correlated with NP concentration. However, in the current study, we did not observe any significant difference in shoot and root biomass of S. nigrum post-PS bead treatment. This contrasting result might be attributed to the fact that, unlike previous studies, which focused on crop plants, our study was focused on a non-crop plant (Liu et al., 2013; Pignattelli, Broccoli & Renzi, 2020). S. nigrum is a strong resistant plant compared to the crop plants. S. nigrum can better adapt to a contaminated environment than crops, and the toxicity of MPs has less effect on its growth performance.

MPs do not easily traverse the barrier (cell wall) of intact plant tissues (Fahn, 1982). Previous studies have shown that metals, oxides, metal nanoparticles, and carbon nanoparticles can cross these barriers and be absorbed by the plant cells (Cunningham et al., 2018; Zhu et al., 2021). Therefore, many reports focused on the apoplastic uptake and transport of NPs by plants (Sun et al., 2020; Taylor et al., 2020). Zhou et al. (2020) confirmed that PS-MPs and PS-NPs accumulated in different tissues of crop plants following their uptake via roots. In another study, small-sized PS-NPs (38.3 nm) were found in the root tissues of wheat, but not the large-sized PS-NPs (191.2 nm) beads (Zhu et al., 2022). In line with these studies, in the current study, we observed an accumulation of 0.2-µm MPs in intercellular spaces of root tissues of S. nigrum (Figs. 2 and 3). We also observed the distribution of PS-NPs in both stems and leaves, transferred via the vascular system from the roots. These findings suggested that small-sized PS beads could enter plant cells more easily, resulting in a higher absorptivity in tissues (Li et al., 2023b). The PS beads were mainly distributed in the epidermis and xylem tissues of S. nigrum roots in our study; these results were consistent with those of previous studies (Li et al., 2019b; Liu et al., 2022). Future studies should focus on the further elucidation of the effects of MPs on plant growth, reproduction, and accumulation. This data might prove crucial in assessing the potential risks of MPs to food safety and environmental sustainability.

Supplemental Information

Supplemental Information 1 Structure diagram of root (A), stem (B) and leaf (C)

Click here for additional data file.

Supplemental Information 2 Effect of exposure to NB-labeled PS-MPs beads on the plant height, shoot and root biomass of S. nigrum

Plant heights were 15.6 cm and 15.0 cm after treatment with 0 and 0.2 µm MPs, respectively. The presence of PS beads had no significant effect on plant height and root and shoot biomass of S. nigrum seedlings after exposure to PS beads for 30 d.

Click here for additional data file.

Additional Information and Declarations

Competing Interests

Author Contributions

Data Availability

The authors declare there are no competing interests.

JuKui Zhang conceived and designed the experiments, performed the experiments, analyzed the data, prepared figures and/or tables, authored or reviewed drafts of the article, and approved the final draft.

Lian Cao performed the experiments, analyzed the data, prepared figures and/or tables, and approved the final draft.

Xiaoyan Zhu analyzed the data, authored or reviewed drafts of the article, and approved the final draft.

Hanbo Li analyzed the data, prepared figures and/or tables, and approved the final draft.

Gang Duan performed the experiments, prepared figures and/or tables, and approved the final draft.

Ying Wang conceived and designed the experiments, performed the experiments, analyzed the data, prepared figures and/or tables, authored or reviewed drafts of the article, and approved the final draft.

The following information was supplied regarding data availability:

The raw measurements are available in the Supplementary File.

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
