# Peer review of "Accumulation and transfer of polystyrene microplastics in Solanum nigrum seedlings"

_PeerJ, doi:10.7717/peerj.15967_

## Round 0.1 · original submission · Minor Revisions

Dear authors both reviewers and I found your work interesting. Please have a look at their comments and answer all of them. Concerning the changes proposed by reviewer 1 on substitution of some references: Please check carefully if those suggestions are correct and fix those that do not refer to your statements/results. You can add some of the suggested references and keep also those that you consider appropriate.

Reviewer 1 ·

Basic reporting

The authors studied the Accumulation and transfer of microplastics in Solanum nigrum seedlings. Although the experimental design is simple, the study provide some interesting findings. I recommend major revisions before considering its acceptance.
Title: Only PS MPs were studied, which should be clarified in the title.
L32: add space in “andweretransferred”; check the whole manuscript.
L51: “Kv et al., 2022“” was not found in references.
L56-57: The effects of MPs in soil-plant systems have been well summarized in several reviews, which may be better to support your statement. I suggest replace (Jiang et al., 2019; Bouaicha et al., 2022a) with the reviews below:
Wang, F., Feng, X., Liu, Y., Adams, C. A., Sun, Y., & Zhang, S. (2022). Micro (nano) plastics and terrestrial plants: Up-to-date knowledge on uptake, translocation, and phytotoxicity. Resources, Conservation and Recycling, 185, 106503.
Wang, F., Wang, Q., Adams, C. A., Sun, Y., & Zhang, S. (2022). Effects of microplastics on soil properties: current knowledge and future perspectives. Journal of Hazardous Materials, 424, 127531.

L57-59: The recitations need to be updated. Here you only cite a reference on the Accumulation of plastics in aquatic food webs. What about plants and soil animals? I recommend two references. Resources, Conservation and Recycling, 2022, 185, 106503. Wang, et al. (2022). Interactions between microplastics and soil fauna: a critical review. Critical Reviews in Environmental Science and Technology, 52(18), 3211-3243.

L69: “Maity et al., 2021” is a wrong citation, which focused on “Functional interplay between plastic polymers and microbes”.
L68-69: there have been many studies on plant uptake of MPs, see Review: Resources, Conservation and Recycling, 2022, 185, 106503.
L72: remove “,”
L162-163: (Huffer, Weniger & aHofmann et al., 2018) is a wrong citation, which studied “Sorption of organic compounds by aged polystyrene”. Also check Tourinho et al., 2019
L164: Lian et al., 2020 “found that the presence of PSNPs could partially reduce Cd contents in leaves and alleviate Cd toxicity to wheat”. Please check if this reference support your opinion.
L166: there are two references by Li et al., in 2019 (L241, L249), which one is cited here? You may mark them as 2019a and 2019b.
L161-167: this paragraph tells readers why you conduct this study, which can be moved to Introduction.
L231: Advanced Science, 9(33), 2202336.
L263: correct “cd” as “Cd”

Experimental design

Simple, but acceptable.

Validity of the findings

New and interesing.

Additional comments

major revisions.

Reviewer 2 ·

Basic reporting

Zhang et al. presents a work entitled “Accumulation and transfer of microplastics in Solanum nigrum seedlings” in which they investigated the transference of polystyrene beads (microplastics) from soil to plant tissues. The results were obtained by confocal laser scanning and scanning electron microscopy of the plant tissues.

The article is short and concise. The writing is clear, unambiguous, and technically correct. The structure of the article has an acceptable format. Introduction has sufficient background with novel literature.

The figures and tables should be improved. Some recommendations to take into account are: 1) add the number of replicates used when applicable in the table/figure caption; 2) add the size of scale bars when applicable in the figure caption; 3) in figures with images add the microscopy technique and observation conditions in the figure caption, each figure should be understood without the rest of the manuscript; 4) in figures with images add legends (with arrows if necessary) to indicate the different parts/tissues of the plant sample shown, it could be a useful piece of information; 5) I also recommend adding at the top of each column a brief identifying name for each set of results (in figures with images).

Experimental design

Research objective is clear in lines 78-80. The methods are well described. I think that the experimental design is acceptable, however I am concerned about the limited number of biological replicates.

Validity of the findings

Observing the green fluorescence was difficult for me (Figs. 4-6). I recommend increasing the brightness to facilitate this task. Please indicate in the figure caption if you modify the brightness and save the native files of these images in the supplementary material.

The discussion is of sufficient quality with novel literature.

Additional comments

A possible typo was detected:
- Line 77 “isused toremediate” -> “isused to remediate”
- Line 165 “S. solanum is a” -> “S. nigrum is a”
- Line 177 “S. nigrumis a strong” -> “S. nigrum is a strong”

---

## Round 0.2 · Minor Revisions

Dear author can you check the problems with the cited references indicated by reviewer 1? Thanks

Reviewer 1 ·

Basic reporting

There are still wrong citations. The authors should carefully check the citations one by one co confirm they match the cited references.

L69: Ming, 2009 is wrong citation. Provide the detailed information of this reference (L268-269). Find the paper and read it.
Avellan et al., 2017 is also a wrong citation, which studied gold nanoparticles, not microplastics.

Experimental design

Good.

Validity of the findings

Good

Reviewer 2 ·

Basic reporting

The authors have responded acceptably to my comments. The main problems with the manuscript have been resolved. The results are interesting and, if the editor or another reviewer has no objections, suitable for publication.

Experimental design

Not applicable

Validity of the findings

Not applicable

Additional comments

Not applicable

---

## Round 0.3 · accepted · Accept

Dear authors, your manuscript is now ready for publication. Thanks for taking into account all reviewers comments and suggestions.